# Exploring the Effects of Industrial Land Transfer on Urban Air Quality Using a Geographically and Temporally Weighted Regression Model

**DOI:** 10.3390/ijerph20010384

**Published:** 2022-12-26

**Authors:** Lan Song, Zhiji Huang

**Affiliations:** 1School of Public Administration and Policy, Renmin University of China, Beijing 100872, China; 2School of Government, Central University of Finance and Economics, Beijing 100081, China

**Keywords:** industrial land, land transfer, air pollution, air quality index, geographically and temporally weighted regression model

## Abstract

This paper explores the spatial-temporal heterogeneity of the impact of industrial land transfer on urban air quality using the air quality index (AQI) and primary land market transaction data of 284 cities from 2015 to 2019 in China. Based on a three-dimensional conceptual framework including scale, price and style effect of industrial land transfer, we find that: (1) The scale effect shows an obvious characteristic of spatial agglomeration, and the agglomerations transfer from central and northern China to the western and southeast coastal regions. (2) Industrial land transfer price has a greater impact on air quality than transfer scale no matter whether the effect is positive or negative, which may be because the expansion scale of construction land is restricted strictly by indicators. (3) The scale of industrial land transferred by agreement in the west and northeast will reduce the air quality. (4) The impact of industrial land price transferred by bidding, auction and listing on AQI is gradually decreasing, but that of land transferred by agreement is still high in the northwest and northeast regions. Finally, we put forward policy recommendations based on the spatial and temporal heterogeneity of these effects, which will help alleviate or avoid environmental problems caused by land resources mismatch and industrial development.

## 1. Introduction

Over the past few decades, China has experienced rapid urbanization and industrialization [1]. However, such exceptional economic performance has come at the cost of environmental degradation [2,3]. The 2022 Environmental Performance Index (EPI), jointly released by Yale Center for Environmental Law and Policy and the Center for International Earth Science Information Network at Columbia University, showed that China’s EPI score ranked only 160th out of 180 countries [4]. Air pollution in China is particularly severe and has become a global concern [5,6,7]. As the largest developing country in the world, China has adopted green and low-carbon approaches in its economic and social development and worked to build a modernized country in which humanity and nature coexist in harmony [8]. The realization of this transition goal would not only improve the living environment and economic development quality but also provide experience for other developing countries in transition development. However, the transition process is quite difficult and endless.

In urban China, land is a strategic tool for economic development [9]. In addition, the land supply of local governments is closely related to air pollutant emission. Under the dual incentives of “land finance” and “land politics”, local governments strategically supply land [10,11,12,13]. First is the structural strategy, namely competing to expand the scale of industrial land transfers and reducing transfer prices to attract manufacturing investment to stimulate local industrial economic growth, which increases the price of commercial and residential land to maximize the extra budgetary fiscal revenue. Second is the intercity strategy. Manufacturing investments are limited and mobile. So local governments have to race to offer more low-priced industrial land to attract investments, which forms a strategic interaction between local governments in industrial land supply and makes the transfer price spatially relevant [14]. In addition, Wang et al., proposed that the strategic interaction of local governments’ industrial land supply is featured by ‘strengthening intervention with a complete race-to-the-bottom strategy’ [15].

Under the intervention of local governments, land resource allocation is distorted and has brought significant negative impact on air quality [16,17,18,19]. However, most studies agree that the effect is stable in both a temporal and spatial dimension. In fact, the development level among Chinese cities is different. At different development stages, industrial expansion has different relations with the environment, which has been proved by the environmental Kuznets curve [20,21,22]. So, it is worth exploring the spatial and temporal heterogeneity of the impact of industrial land on urban air quality. In view of this, based on the framework of Chen et al.—i.e., scale effect, price effect, style effect [19]—we took the non-stationarity of space and time into account and employed the geographically and temporally weighted regression model (GTWR) to quantify these three effects. In addition, we chose Air Quality Index (AQI) rather than carbon emission data as a proxy variable for air quality, which would provide new empirical data support for relevant research.

## 2. Literature Review

Some literature has studied the direct impact of urbanization and industrialization on air quality. Among them, carbon emission is the most commonly used proxy variable for air quality. Li and Lin proved that industrialization tends to increase carbon dioxide emissions [23]. Tu found that every one percentage point increase in economic scale will result in an average increase of 15 Mt in carbon emissions and every one percentage point rise in the share of manufacturing industry produces an average of 56 Mt carbon emissions [24]. From the perspective of heterogeneity, this impact varies across regions, time and development stages. Wang et al. found that the impact of industry structure on air pollutants in underdeveloped regions is higher than in developing and highly developed regions due to pollution industries transfer [25]. Phetkeo Poumanyvong found that urbanization elasticity of emissions in the middle-income group is larger than that in low and high groups [26]. With the continuous advance of industry transfer, pollution transfers are caused by pollution-intensive industry transfer, which intensifies the environmental pressure in the central and the western region of China [27,28].

Some literature discussed the internal mechanism of industrialization affecting urban air quality. First is the change of land use pattern. Built-up land occupied cropland is the main land use transfer type [29]. Build-up land use patterns cause higher air pollutant concentration than agriculture and forest land use [30]. Moreover, urban fragmentation significantly affects air quality and the concentration of air pollutants [31]. Second is land use efficiency. Land use efficiency integrally reflects the degree of material circulation and energy exchange [32]. Achieving “green” use of industrial land is beneficial to save precious land resources and protect ecological environments [33]. Innovation-oriented land use transition is also conducive to reducing industrial pollution emissions [34].

In China, many research projects focus on the environmental effect of land resource mismatch caused by government intervention. The influencing mechanism can be summarized into three aspects. First, industrial enterprises that obtain land at a lower price through a non-market approach are usually enterprises with poor quality and serious pollution [35,36,37]. Second, the mismatch of land resources leads to the distorted allocation of other production factors, which reduces the productivity of enterprises and is not conducive to technological innovation, thus bringing more environmental pollution [38,39,40]. The third is the pollution transfer caused by industrial transfer, which is explained by the pollution haven hypothesis [27,41].

The above research papers reveal the general impact of local government land transfer on air quality, and some of them have already noticed the heterogeneity even though they need to be deepened. For example, Sun et al. found that with the marketization of the industrial land conveyance price, urban industrial pollution is presenting an inverted U-shaped change trend, and the inflection point varies in different cities [42]. To this, the purpose of this study is to explore the spatial and temporal heterogeneity of the impact of industrial land transfer on air quality and reveal the influence of local government intervention on it. The research results are helpful to understand and realize high-quality development from the perspective of industrial land transfer.

## 3. Methodology

### 3.1. Data Source

The dependent variable in the model is the average annual air quality index (AQI) from 2015 to 2019, which is the arithmetic mean of daily AQI in each year. AQI is a comprehensive evaluation of six major pollutants of SO_2_, NO_2_, PM10, PM2.5, CO, and O_3_ and accurately reflects air quality. The higher the AQI, the worse the air quality. Air quality data used in this article come from the China Environmental Monitoring Center (http://www.cnemc.cn/, accessed on 12 May 2022).

The core explanatory variables are the land transfer scale and price, which is collected from the China Land Market Network (https://www.landchina.com/, accessed on 12 May 2022). During the observation period, we collected 1,296,376 land transaction deals. Then we cleaned and summarized the information of each land transaction at the city level. Specifically, it is divided into the total transfer scale (SCALE) and price (PRICE), transfer scale (ZPG_SCALE) and price (ZPG_PRICE) by bidding, auction and listing, transfer scale (AGRE_SCALE) and price (AGRE_PRICE) by agreement. The construction period of most industrial projects is two years, so previous studies included the current, one-period and two-period lagged variables of land variables in a model [35,36,37]. Considering the possible situation that the construction period is not completed within two years and the construction period is more than two years, this paper also added three-period lagged variables. They are successively denoted by SCALE_*n*, PRICE_*n*, ZPGSCA_*n*, ZPGPRI_*n*, AGRSCA_*n*, AGRPRI_*n*, *n* denotes lag period. If *n* = 1, it represents the first-order lag variable, the second-order lag variable for *n* = 2 and the third-order lag variable for *n* = 3. Land transfer variables have three lag periods, so their observed time is from 2012 to 2016.

The selection of control variables incorporates as many factors as possible based on previous studies, including ① Population density (DEN); ② The proportion of tertiary industry in GDP (TER). Because advanced industrial structure means improvement of production technology and decrease of energy consumption, the proportion of the tertiary industry is used to control the impact of the industrial structure on air pollution; ③ Per capita GDP (PGDP), used to control the level of economic development; ④ The proportion of total import and export trade in GDP (FTD). The level of foreign trade transactions is closely related to the type of industry and economic level, so it is necessary to control the degree of dependence on foreign trade; ⑤ Financial pressure (FIN), which is calculated by subtracting fiscal revenue from fiscal expenditure and dividing it by fiscal revenue. Land finance is one of the main motives for local governments to transfer land, so it is necessary to control the financial situation of cities; ⑥ Number of environmental penalties (PUN), a variable that reflects a city’s attention on environmental governance, which will improve the urban environment, so it is used to control the social investment intensity of pollution governance; ⑦ The average annual rainfall (RAIN); ⑧ Average annual temperature (TEMP); ⑨ Average annual wind speed (WIND); these three variables are natural factors that have an important effect on atmospheric movement, and they also affect the monitoring value of air quality index. So, it is necessary to control and eliminate the impact of natural factors on air quality. Statistics of environmental administrative punishment are from the Magic Weapon of Peking University (https://www.pkulaw.com/, accessed on 12 May 2022). Meteorological data are collected from the National Climatic Data Center of the United States (https://www.ncdc.noaa.gov/, accessed on 12 May 2022). The other control variables required for the study are all derived from the China Urban Statistical Yearbook. DEN, TER, PGDP, FTD are two-period lagged in order to prevent the possible endogenous risks between the control variables and the core explanatory variables, so they are observed from 2013 to 2017.

Due to the partial absence of land and socio-economic data, the final empirical sample included the panel data of 284 cities at prefecture level or above. Table 1 is the description of variables.

### 3.2. Conceptual Framework

Chen et al. measured the effects of industrial land supply on air quality from scale, method and price dimension respectively [19]. Based on their analysis framework, we more specifically summarized and discussed the spatiotemporal heterogeneity of the impact of the industrial land transfer on air quality.
Price effect;

Land prices have a direct impact on land use efficiency and sustainable development. Unreasonable industrial land price not only leads to a waste of resources but also decreases land use efficiency and increases production costs [43]. Especially, pollution-intensive industries are more sensitive to land costs and tend to locate where the industrial land price is low [41]. It is found that the greater the price difference between commercial and industrial land, the deeper the mismatch of land resources. As a result, industry development is excessive and tertiary industry development is restrained, which increases environmental pollution [39].
Scale effect;

On the one hand, massive urban land expansion is accompanied by increased environmental pollution [24]. In China, a large part of urban land expansion has been carried out for industrial use. The expansion of urban industrial land transfer significantly reduces urban air quality, and this effect is impacted by the transfer style [44]. On the other hand, land transfer scale is closely related to land price. With the marketization of industrial land, land price is improving, stimulating local governments to increase the industrial land transfer scales to maximize land transfer profits. To rapidly develop the economy, the expansion of the industrial scale has been mainly dominated by heavy industries with higher taxes, which will further worsen the urban environment [42].
Style effect;

Due to the immature primary land market, industrial land transactions are not completely realized through market mechanism. Before the reform and opening up, China’s land resources were obtained through administrative allocation and free of charge (called “Huabo” in Chinese), which was suited to the planned economic system. In 1990, the State Council enacted *Interim Regulations Concerning the conveyance and Transfer of the Right to the Use of State-Owned land in Urban Areas*, which marked the establishment of the urban land market system [13]. However, the degree of marketization of land transfer was relatively low for a long time. Many transactions were realized through agreement (called “Xieyi” in Chinese), which was similar to one-to-one negotiation between land user and local government. In 2006, *Notice on the Issues Regarding Strengthen Land Macro-Control*, issued by the State Council, stipulated that industrial land was ordered to be transferred through bidding, auction or listing (collectively called “Zhaopaigua” in Chinese) and the land price is not allowed to be lower than the benchmark price. Meanwhile, the local government was ordered to promulgate the land transfer plan in advance and transfer results after it was completed on China’s land market network (www.landchina.com, accessed on 12 May 2022). Since then, the standardization of state-owned land transfer behavior has been greatly improved. However, industrial land price was still low, and the gap between industrial land and commercial or housing land price was large [45].

On the one hand, the style of agreement transfer avoids market competition, which implies the introduction of low-quality investment projects, and harms both economic development and environmental quality [36,37]. On the other hand, the style of agreement transfer lowers the entry threshold for industrial enterprises and is not conducive to forcing enterprises to innovate, which impedes industrial upgrading and environmental improvement. The larger the area or proportion of industrial land transferred by agreement is, the higher the emissions of industrial pollutants are [36,37,46]. The closer the agreement transfer price is to the lowest supply price of industrial land, the industrial energy carbon emissions are higher [16].

### 3.3. Spatial Analysis with GTWR Model

In order to reveal the heterogeneity and spatiotemporal evolution characteristics of the impact of land transfer on air quality, this study uses a geographically and temporally weighted regression model (GTWR). Huang et al., first proposed the GTWR model, which brings a time factor into a GWR model so that it can deal with spatial and temporal non-stationarity at the same time [47]. The model can be expressed as follows:(1)$$Y_{i}=\beta_{0}(u_{i},v_{i},t_{i})+\underset{k}{\sum }\beta_{k}\left(u_{i},v_{i},t_{i}\right)X_{ik}+\varepsilon_{i}$$
where ($u_{i},v_{i},t_{i}$) denotes the space-time location of the point *i*, $\beta_{0}(u_{i},v_{i},t_{i})$ represents the intercept value, and $\beta_{k}\left(u_{i},v_{i},t_{i}\right)$ is a set of values of parameters at point *i*. The problem here is to provide estimates of $\beta_{k}\left(u_{i},v_{i},t_{i}\right)$, for each variable *k* and each space–time location *i* ([47]). The estimation of $\beta_{k}\left(u_{i},v_{i},t_{i}\right)$ can be expressed as follows:(2)$$\hat{\beta }\left(u_{i},v_{i},t_{i}\right)={\left[X^{T}W\left(u_{i},v_{i},t_{i}\right)X\right]}^{-1}X^{T}W\left(u_{i},v_{i},t_{i}\right)Y$$
where $W\left(u_{i},v_{i},t_{i}\right)$ = diag($\alpha_{i1}$, $\alpha_{i2}$, …, $\alpha_{in}$) and *n* is the number of observations. Here the diagonal elements $\alpha_{ij}$(1 ≤ *j* ≤ *n*) are space–time distance functions of (*u, v, t*) corresponding to the weights when calibrating a weighted regression adjacent to observation point *i*. To calibrate the model, it is assumed that the observed datapoints ‘close’ to point *i* in the space–time co-ordinate system has a greater influence in the estimation of the $\beta_{k}\left(u_{i},v_{i},t_{i}\right)$ parameters than the data located farther from observation *i*. Hence, the key problem of the GTWR model is defining and measuring the closeness in a space-time co-ordinate system. Considering that location and time are usually measured in different units and have different scale effects, it is more appropriate to use an ellipsoidal co-ordinate system to measure the ‘closeness’ between a regression point and its surrounding observed points.

Given a spatial distance $d^{S}$ and a temporal distance $d^{T}$, then combine them to form a spatial-temporal distance $d^{ST}$ such that
(3)$$d^{ST}=d^{S}⊗d^{T}$$
where ⊗ can represent different operators. If the ‘+’ operator is adopted to measure the total spatio-temporal distance $d^{ST}$, then it is expressed as a linear combination between $d^{S}$ and $d^{T}$.
(4)$$d^{ST}=\lambda d^{S}+\mu d^{T}$$
where λ and μ are scale factors to balance the different effects used to measure the spatial and temporal distance in their respective metric systems.

Following Equation (4), if the Euclidean distance and Gaussian distance–decay-based functions are used to construct a spatial–temporal weight matrix, we will have
(5)$${\left(d_{ij}^{ST}\right)}^{2}=\lambda [{\left(u_{i}-u_{j}\right)}^{2}+{\left(v_{i}-v_{j}\right)}^{2}]+\mu {\left(t_{i}-t_{j}\right)}^{2}$$
(6)$$\begin{array}{l} \alpha_{ij}=exp\left\{-\left(\frac{\lambda [{\left(u_{i}-u_{j}\right)}^{2}+{\left(v_{i}-v_{j}\right)}^{2}]+\mu {\left(t_{i}-t_{j}\right)}^{2}}{h_{ST}^{2}}\right)\right\}\\ =exp\left\{-\left(\frac{{\left(u_{i}-u_{j}\right)}^{2}+{\left(v_{i}-v_{j}\right)}^{2}}{h_{S}^{2}}+\frac{{\left(t_{i}-t_{j}\right)}^{2}}{h_{T}^{2}}\right)\right\}\\ =exp\left\{-\left(\frac{{\left(d_{ij}^{S}\right)}^{2}}{h_{S}^{2}}+\frac{{\left(d_{ij}^{T}\right)}^{2}}{h_{T}^{2}}\right)\right\}\\ =exp\left\{-\frac{{\left(d_{ij}^{S}\right)}^{2}}{h_{S}^{2}}\right\}\times exp\left\{-\frac{{\left(d_{ij}^{T}\right)}^{2}}{h_{S}^{T}}\right\}\\ =\alpha_{ij}^{S}\times \alpha_{ij}^{T} \end{array}$$
where $\alpha_{ij}^{S}=exp\left\{-\frac{{\left(d_{ij}^{S}\right)}^{2}}{h_{S}^{2}}\right\}$, $\alpha_{ij}^{T}=exp\left\{-\frac{{\left(d_{ij}^{T}\right)}^{2}}{h_{S}^{T}}\right\}$, ${\left(d_{ij}^{S}\right)}^{2}={\left(u_{i}-u_{j}\right)}^{2}+{\left(v_{i}-v_{j}\right)}^{2}$, ${\left(d_{ij}^{T}\right)}^{2}={\left(t_{i}-t_{j}\right)}^{2}$, $h_{ST}$ is a parameter of spatio-temporal bandwidth, and $h_{S}^{2}=h_{ST}^{2}/\lambda$ and $h_{T}^{2}=h_{ST}^{2}/\mu$ are parameters of the spatial and temporal bandwidths, respectively.

Let *τ* denote the parameter ratio μ/λ and λ≠0, we can rewrite Equation (5) by normalizing the coefficient of $d_{S}$:(7)$$\frac{{\left(d_{ij}^{ST}\right)}^{2}}{\lambda }=[{\left(u_{i}-u_{j}\right)}^{2}+{\left(v_{i}-v_{j}\right)}^{2}]+\tau {\left(t_{i}-t_{j}\right)}^{2}$$

Without loss of generality, we set λ = 1 to reduce the number of parameters in practice, and so only *μ* has to be determined. In this study, *μ* can be optimized using cross-validation in terms of R^2^ or AIC if no a priori knowledge is available.

In this paper, the GTWR model can be specifically expressed as:
$$\begin{array}{l} lnAQI_{i}=\beta_{0}(u_{i},v_{i},t_{i})+\beta_{1}\left(u_{i},v_{i},t_{i}\right)lnLand\_1_{i}+\beta_{2}\left(u_{i},v_{i},t_{i}\right)lnLand\_2_{i}+\beta_{3}\left(u_{i},v_{i},t_{i}\right)lnLand\_3_{i}\\ \;+\beta_{4}\left(u_{i},v_{i},t_{i}\right)lnDEN\_2_{i}+\beta_{5}\left(u_{i},v_{i},t_{i}\right)lnTER\_2_{i}+\beta_{6}\left(u_{i},v_{i},t_{i}\right)lnPGDP\_2_{i}\\ \;+\beta_{7}\left(u_{i},v_{i},t_{i}\right)lnFTD\_2_{i}\\ \;+\beta_{8}\left(u_{i},v_{i},t_{i}\right)FIN_{i}+\beta_{9}\left(u_{i},v_{i},t_{i}\right)lnPUN_{i}+\beta_{10}\left(u_{i},v_{i},t_{i}\right)lnRAIN_{i}+\beta_{11}\left(u_{i},v_{i},t_{i}\right)lnTEMP_{i}\\ \;+\beta_{12}\left(u_{i},v_{i},t_{i}\right)lnWIND_{i}+\varepsilon_{i} \end{array}$$

To remove the heteroscedasticity effect, we conducted logarithmic treatment of the variables data. Here “Land” is a collection of six land variables. For example, when the land variable is SCALE, $\beta_{1}$, $\beta_{2}$ and $\beta_{3}$ are the scale effects of the first-order lag, second-order lag and third-order lag respectively. More environmental effects to be verified and their corresponding land variables are shown in Figure 1. FIN has negative values and cannot be logarithmic, so we divided it by 100. There are many zero values in land transfer data, so we add 1 to the original data before taking the logarithm in order to prevent data loss. The GTWR models must be applied to variables with a low correlation, so we diagnosed multicollinearity by the variance inflation factor (VIF) before taking the geographically weighted regression. Generally, a VIF value of no more than 7.5 ensures there is no multicollinearity and redundant independent variables in the regression model [48]. The test results show that all VIF values of regression variables are less than 5, which implies there is no multicollinearity problem and conforms to one of the preconditions of the GTWR model.

## 4. Results

### 4.1. Estimation Results of Main Parameters

In this section, we conducted six panel data regressions. Table 2 reports the estimation parameters of the six models, including bandwidth, standard deviation of residuals, residual sum of squares, AIC and R^2^. The optimal bandwidth obtained by cross validation is between 0.352 and 0.382. In addition, compared with the OLS model, R^2^ increases significantly to about 0.98. This exactly implies that temporal and spatial non-stationary effects are much significant, and consequently GTWR model achieves a better goodness-of-fit than that of the OLS model in terms of R^2^.

### 4.2. Descriptive Analysis of t-Value and Coefficient

Table 3 reports significance (*t*-value) and Table 4 reports regression coefficients of land explanatory variables. There are 1420 samples in total, and the number of samples with significant regression coefficients of land explanatory variables accounts for about 34–48%, indicating that land transfer will significantly affect urban air quality in these samples. Moreover, the number of samples that are significant at a 99% confidence level is the most, followed by 95%, and the number of samples that are significant at a 90% level is the least. So the overall significance of these samples is at a high confidence level.

From the perspective of regression coefficients (Table 4), the regression coefficients on the 25th percentile are negative, and those on the 75th percentile are positive, indicating that land transfer has both positive and negative effects on air quality. Except for PRICE_2 and ZPGPRI_2, the mean and median of regression coefficients of other variables are positive. Therefore, it is inferred that the overall effect of land transfer on AQI is positive, which will improve the AQI and deteriorate air quality. In addition, coefficients of land transferred by agreement are smaller than that of transferred by bidding, auction and listing whatever explanatory variables are denoted by area or price.

Table 5 reports significance (*t*-value) and regression coefficients of control variables. Limited by space, we only show two of the six regression results here. Generally speaking, the regression coefficients of control variables are half positive and half negative, which are also heterogeneous. Moreover, the number of control variables‘ coefficients with high significance is more than 50%, which is higher than that of land variables.

### 4.3. Spatiotemporal Evolution Analysis Based on Geographical Visualization

#### 4.3.1. Spatiotemporal Analysis of Scale Effect

Given the limited space available, we choose the coefficients of third lag term of land variables, namely $\hat{\beta_{3}}$ for geographic visualization analysis. The legend adopts the Natural breaks classification, which can place clustered values in the same class. However, it may classify the values near zero as the same class, so that we cannot distinguish whether the effect is positive or negative. Therefore, the following analysis focuses more on the regions with the strongest positive or negative effects. Figure 2 shows the space-time evolution characteristics of the scale effect. First, most coefficients are positive, indicating that the industrial land transfer scale in most cities will aggravate air pollution. Second, the absolute values of regression coefficients decrease slightly from 2015 to 2019, indicating that the scale effect of industrial land transfer may be gradually weakened. Third, from 2015 to 2019, there is an obvious spatial agglomeration of scale effect, and the agglomeration has transferred. In 2015, the values of the scale effect are geographically dispersed. However, since 2017, red and blue color blocks are separately more concentrated. This phenomenon implies that industrial transfer and agglomeration may lead to the spatial agglomeration of scale effect. In addition, the regions where high regression coefficients were collected shifted from Beijing-Tianjin-Hebei and Wuhan (the capital city of Hubei Province) urban agglomeration in 2015 to Yangtze River Delta, northern Guangdong Province, northwestern Gansu Province and Chengdu-Chongqing Economic Circle in 2019. The regression coefficients of Henan, Hebei Province and other central regions are significantly smaller, even from positive to negative, indicating that the transfer of industrial land in these regions will improve urban air quality. Note that before 2017, red-colored cities are mainly distributed in the middle of China, which implies industrial land transfer scale in these cities has more significant positive effects on air quality. However, since 2018, red-colored cities move from the middle to the west and east of China. This result is partly consistent with Hu et al. [49]. They found that pollution-intensive industries are transferred from the east to the west and from the eastern coastal region to the northwest and the middle. However, the agglomeration degree of the positive scale effect in southeast China is still high.

#### 4.3.2. Spatiotemporal Analysis of Price Effect

Figure 3 shows the coefficients spatiotemporal distribution of lnPRICE_3. The marginal effect of industrial land transfer price on AQI is much greater than that of scale, whether it is positive or negative. Why does the price effect be stronger than the scale effect on air quality? Wu and Zhu [50], and Huang and Du [51] have explored the impact of air pollution on industrial land transfer. Both studies use PM2.5 as the independent variable. The difference is that the former uses transfer area and number as the dependent variables while the latter uses transfer price. They find that for every unit reduction in PM2.5, the number of industrial land transfers decreased by about 0.3%, while land prices increase by about 1.6%. Ignoring the estimation bias caused by different control variables and observation time in the two studies, the huge difference between the two-impact effect indicates that it is reasonable that the impact of price is greater than that of area, just as the results of this paper. The institutional explanation behind this phenomenon is that the total land supply of local governments is constrained by the indicators of the central government [52]. As prices are affected by macroeconomic fluctuations, the price effect changes more over time and has greater variance. For example, the values of price effect in 2015 is from −0.40 to 0.62, and up to −0.59–0.99 in 2018, both positive and negative effects are expanded by about 50%. Compared with the scale effect, the price effect has no obvious spatial agglomeration or transfer phenomenon from 2015 to 2019. Henan Province in Central China and Heilongjiang Province in Northeast China are two regions with the highest positive price effect. Yangtze River Delta is always the region with the highest negative price effect, while its scale effect is the highest and positive.

#### 4.3.3. Spatiotemporal Analysis of Style Effect

Scale

In this section, we analyze the style effect by comparing the scale effect and price effect of industrial land transferred in different ways.

As shown in Figure 4, the scale effect of industrial land transferred by agreement is smaller than that by bidding, auction and listing, because the absolute values of its regression coefficients to air quality are smaller. However, according to the previous research results, bidding, auction and listing are more market-oriented transfer approaches, so the use efficiency of industrial land transferred in these ways is generally higher and should cause less air pollution. The contradiction with the previous research results just reflects the existence of non-stationarity in time and space that cannot be ignored. We speculate that with the marketization and standardization of the land transaction system, the proportion of land transferred by agreement has become smaller and smaller, and the quality of industrial production projects attracted has improved, and in some regions, local governments are likely to attract high-end enterprises rather than traditional industries in this way. As a result, the deterioration of urban air quality has been alleviated. These possible reasons also apply to the following explanations of price effect differences.

From the perspective of temporal and spatial change trend, the regions with the highest positive scale effects of land transferred by bidding, auction and listing move to the eastern coastal, northeast and northwest regions from 2015 to 2019, while they move to northeast and northwest regions for land transferred by agreement. Therefore, we conclude that in the southwest and northwest regions, the agreement is still an important land transfer style to attract industries, and tends to aggravate air pollution. In the eastern coastal areas, due to the high degree of marketization, the expansion of industrial land is mainly achieved through bidding, auction and listing, so the agreement transfer will not bring serious air pollution.

Price

Figure 5 shows the coefficients spatiotemporal distribution of lnZPGPRI_3 and lnAGRPRI_3. Similar to the scale effect, the price effect of industrial land transferred by agreement is also smaller than that by bidding, auction and listing. However, the difference in regression coefficient values is far greater than that of scale effect. The price effect of industrial land transferred in different ways also shows different agglomeration characteristics. For style of bidding, auction and listing, the places with high positive price effects are mainly concentrated in the middle in 2015, such as Henan and Hubei Province. The decentralized trend appeared in 2018. In 2019, except for the relatively small high-value agglomeration in the northeast, the price effect in most regions of the country decreased significantly. For the style of agreement, the cities with large price effects were originally widely distributed in the middle and northeast, and then transferred to the northwest and northeast. According to the findings of Wang et al., when the level of economic development is low, local governments tend to attract investment at low land prices [52]. Northwest and northeast China are economically underdeveloped, so cities in these regions are more inclined to supply land at low prices through agreement and lack of incentives to select enterprises. As a result, the industrial land transferred through agreement in these cities will significantly reduce the urban air quality.

## 5. Discussion

This paper analyzes the temporal and spatial heterogeneity of the impact of industrial land transfer on urban air quality. The R^2^ of GTWR model is much higher than that of the OLS model, indicating that land transfer characteristics and their environmental effects differ among cities. Previous work has found that cities with different industrialization stages or economic development levels may face different Kuznets curves [42]. In addition, we clearly show where the differences are and how they change over time.

In reviewing the literature, most papers only studied one of the effects of land transfer on air quality. This paper considers all three effects. Although Chen et al. have discussed these three effects before [19], actually they only discussed the impact of the land transfer scale and the agreement transfer style. In this paper, we conduct six regressions and focus more on comparative analysis of different effects. Therefore, another contribution of this paper is to conduct multi-dimensional research on the impact of industrial land transfer on air quality.

Surprisingly, the price effect is greater than the scale effect. We believe that this is because the indicators of state-owned construction land are severely constrained [52], and their spatiotemporal change trend is relatively stable, so their environmental effects are small. This implies that the related research that only selects the expansion scale of industrial land as the independent variable needs to rethink the potential impact of construction land index constraints on the results.

What is more, the influence range of industrial land transferred by agreement on air quality is smaller than that of industrial land transferred by bidding, auction and listing. This seems to indicate that the agreement transfer style will not cause serious pollution to the air quality, which is inconsistent with previous research [35,36,37]. In fact, this range only represents the difference between the maximum and minimum of the effects of different transfer styles. As far as urban individuals are concerned, it will be found that the industrial land transferred by agreement in economically underdeveloped areas will significantly aggravate urban air pollution, which is consistent with the conclusions of many previous studies.

However, some spatial problems, such as the spatial correlation of error and the autocorrelation of dependent variables, should be better considered to reduce estimation bias. With the market-oriented reform of land transfer, the proportion of agreement transfer continues to reduce, and the intervention channel of local government turns to listing, a land transfer method with considerable discretion for local governments [35]. Therefore, future research can focus on the differences in the environmental effects of these three market-based land transfer methods. It is also worthwhile to seek the inner mechanism of heterogeneity in the future.

## 6. Conclusions

By using the GTWR model, this paper finds significant spatiotemporal heterogeneity between industrial land transfer and urban air quality. First, the scale effect shows an obvious characteristic of spatial agglomeration, and the agglomerations transfer from central and northern China to the western and southeast coastal regions. Second, the land price has a greater impact on air quality than land scale. Considering that the supply of urban construction land is strictly restricted by the central government [52], there is no doubt that local government would be more inclined to attract investment through price strategy. Therefore, price is more likely to be the mechanism of industrial land expansion on urban air quality. Third, considering the influence of transfer style, the scale of industrial land transferred by agreement in the west and northeast will reduce the air quality. This may be attributed to the national strategy of revitalizing of northeastern old industrial base and promoting the development of the central and western regions. These regions are allowed to sell land at a price lower than the “National Bottom Price Standard for Industrial Land Transfer”, resulting in large-scale expansion of industrial land, which has come at the cost of environmental degradation [45]. Fourth, the impact of the price of industrial land transferred by bidding, auction and listing on AQI is gradually decreasing, but the impact of the price of industrial land transferred by agreement on AQI is still high in the northwest and northeast regions, suggesting that these underdeveloped regions may still attract polluting enterprises at low prices.

In view of the above findings, we propose the following policy recommendations: First, in underdeveloped cities, it is recommended that threshold conditions should be set for enterprises planning to obtain land, so as to avoid the agglomeration of high pollution enterprises, which will cause serious deterioration of urban air quality. Second, underdeveloped cities should continue to promote land marketization reform, to give play to the price mechanism and improve the efficiency of industrial land allocation and utilization. Third, for developed cities, local governments should actively explore mixed land use policies to attract investment, and increase policy support for green production and innovation. Fourth, it is recommended that the outflow cities should pay the environmental governance fees for the cities undertaking high pollution industries, so as to achieve the balance between industrial development and environmental governance.

## Figures and Tables

**Figure 1 ijerph-20-00384-f001:**
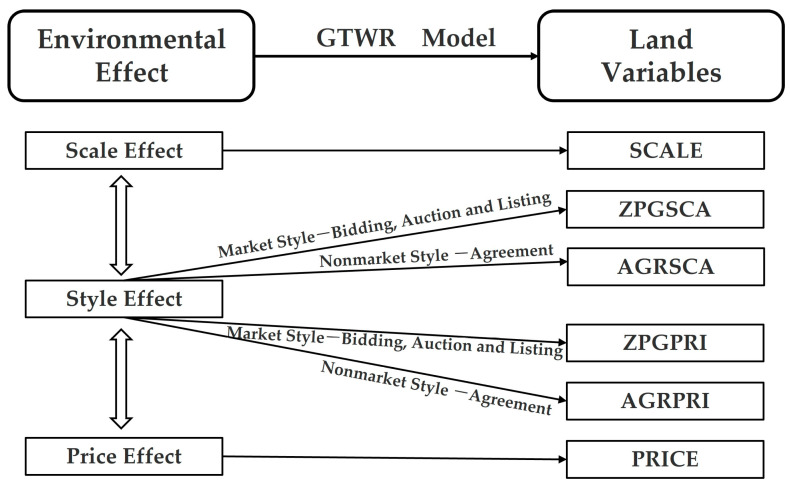
Data processing framework.

**Figure 2 ijerph-20-00384-f002:**
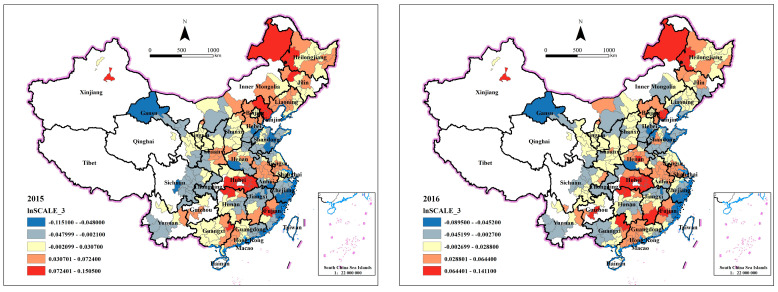
Spatiotemporal distribution of lnSCALE_3’s coefficients.

**Figure 3 ijerph-20-00384-f003:**
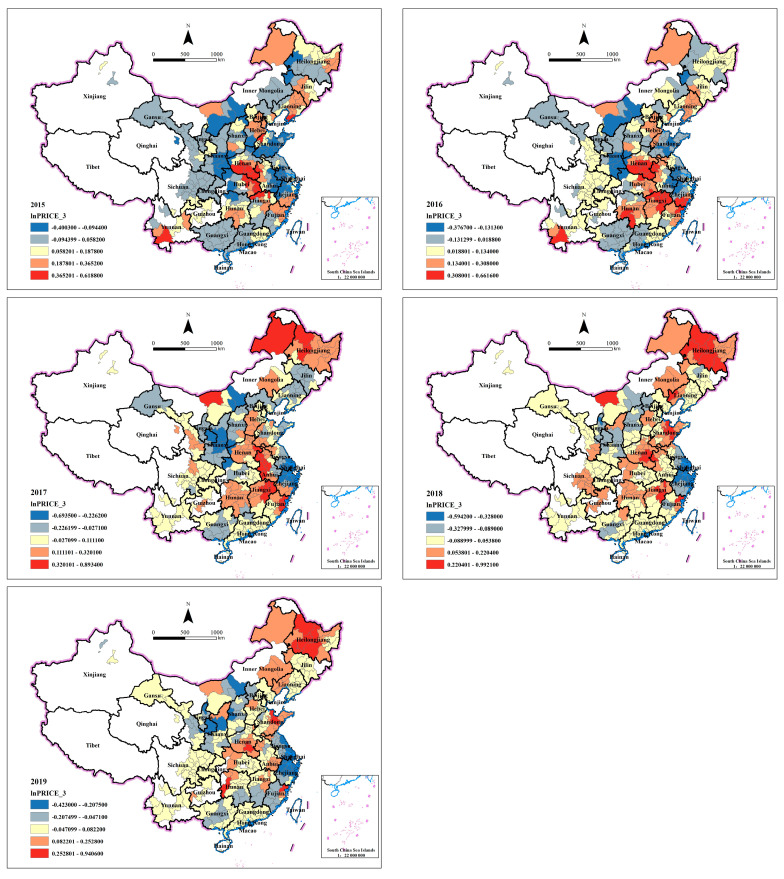
Spatiotemporal distribution of lnPRICE_3’s coefficients.

**Figure 4 ijerph-20-00384-f004:**
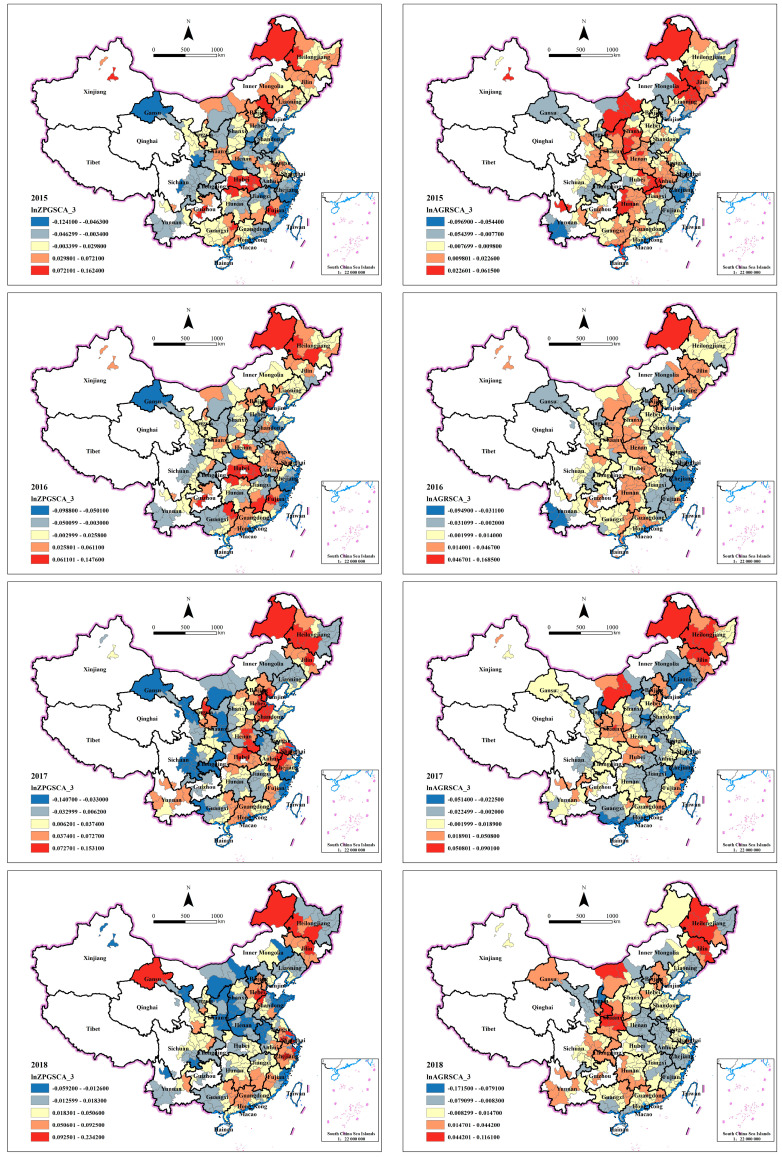
Spatiotemporal distribution of lnZPGSCA_3’s coefficients (**left**) and lnAGRSCA_3’s coefficients (**right**).

**Figure 5 ijerph-20-00384-f005:**
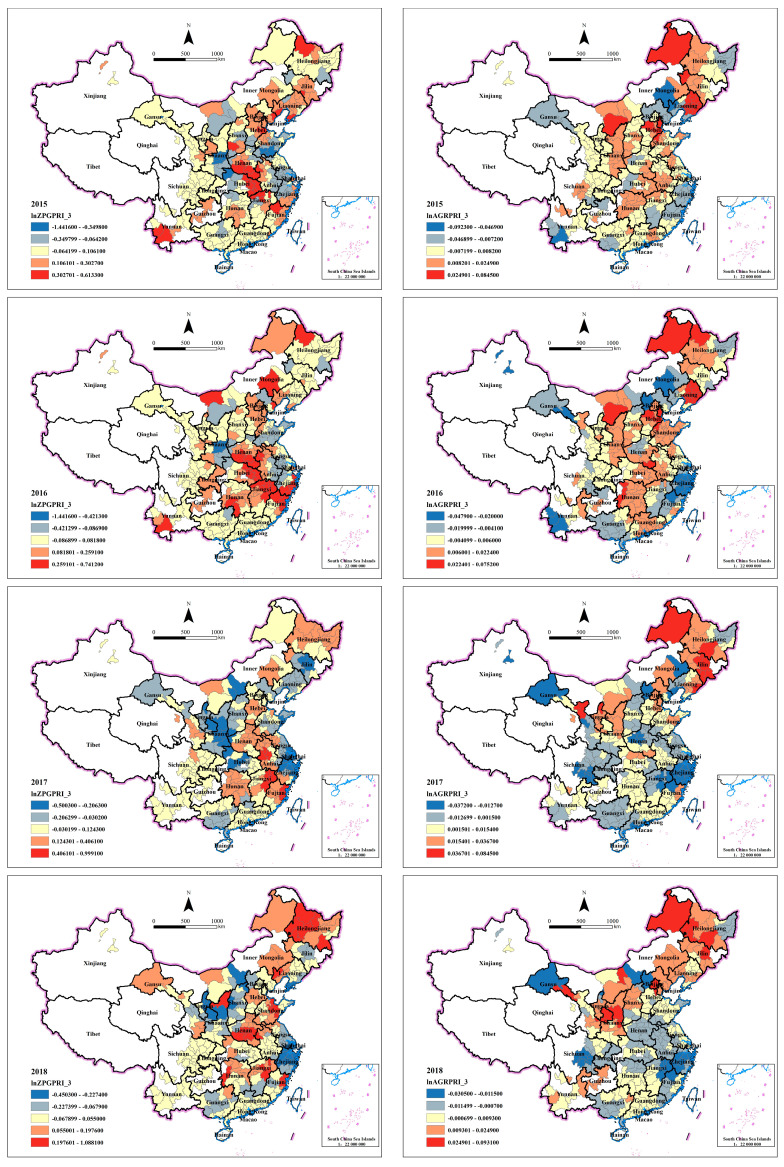
Spatiotemporal distribution of lnZPGPRI_3’s coefficients (**left**) and lnAGRPRI_3’s coefficients (**right**).

**Table 1 ijerph-20-00384-t001:** Description of variables.

	Variable	Mean	Std. Dev.	Min	Max
dependent variable	AQI_avr	78.924	19.629	36.56	146.341
main explanatory variables					
SCALE	SCALE_1	246.371	251.699	0	1932.995
	SCALE_2	270.039	276.234	0	3683.890
	SCALE_3	304.328	307.661	0	3683.890
ZPG_SCALE	ZPGSCA_1	239.464	247.262	0	1932.995
	ZPGSCA_2	260.991	263.440	0	2527.114
	ZPGSCA_3	291.686	290.616	0	2723.786
AGRE_SCALE	AGRSCA_1	6.544	27.427	0	676.119
	AGRSCA_2	8.758	40.504	0	1156.776
	AGRSCA_3	12.495	54.573	0	1183.941
PRICE	PRICE_1	234.823	119.36	0	992.719
	PRICE_2	226.922	113.871	0	766.541
	PRICE_3	218.555	109.105	0	759.222
ZPG_PRICE	ZPGPRI_1	236.759	121.863	0	924
	ZPGPRI_2	230.042	117.914	0	924
	ZPGPRI_3	221.709	113.08	0	924
AGRE_PRICE	AGRPRI_1	112.201	144.279	0	992.719
	AGRPRI_2	118.229	142.311	0	984.888
	AGRPRI_3	120.509	137.015	0	979.008
Control Variables	DEN_2	4.891	3.692	0.015	28.162
	TER_2	47.155	9.965	14.841	79.36
	PGDP_2	52,764.417	35,090.616	8407	467,749
	FTD_2	0.169	0.275	0	2.222
	FIN	175.5151	165.6252	−35.11796	1254.86
	PUN	224.657	589.139	0	8054
	RAIN	1073.085	496.726	203.285	2743.909
	TEMP	14.702	4.483	3.252	24.506
	WIND	2.208	0.486	1.081	4.337

**Table 2 ijerph-20-00384-t002:** Estimation results of main parameters.

	Bandwidth	Std.residuals	SSR	AIC	R^2^	R^2^ (OLS)
SCALE	0.376	0.035	1.752	−9486.342	0.980	0.431
ZPG_SCALE	0.372	0.035	1.692	−9536.100	0.981	0.425
AGRE_SCALE	0.375	0.037	1.957	−9329.737	0.978	0.268
PRICE	0.382	0.038	2.086	−9238.871	0.977	0.236
ZPG_PRICE	0.352	0.033	1.538	−9671.602	0.983	0.241
AGRE_PRICE	0.355	0.031	1.396	−9809.642	0.984	0.246

**Table 3 ijerph-20-00384-t003:** Statistical description of *t*-values of land variables.

	Variable	90%	95%	99%	Sum
SCALE	lnSCALE_1	7.183	13.239	22.465	42.887
	lnSCALE_2	8.310	12.676	24.718	45.704
	lnSCALE_3	7.746	11.831	24.859	44.437
ZPG_SCALE	lnZPGSCA_1	7.394	13.521	24.366	45.282
	lnZPGSCA_2	9.014	12.254	26.408	47.676
	lnZPGSCA_3	7.746	11.690	24.859	44.296
AGRE_SCALE	lnAGRSCA_1	10.423	14.507	17.746	42.676
	lnAGRSCA_2	7.254	12.535	22.817	42.606
	lnAGRSCA_3	8.380	12.254	21.268	41.901
PRICE	lnPRICE_1	8.732	12.183	19.507	40.423
	lnPRICE_2	7.817	10.775	17.676	36.268
	lnPRICE_3	7.465	8.803	23.451	39.718
ZPG_PRICE	lnZPGPRI_1	8.873	11.761	16.620	37.254
	lnZPGPRI__2	8.099	10.141	16.056	34.296
	lnZPGPRI_3	6.479	10.352	19.789	36.620
AGRE_PRICE	lnAGRPRI_1	8.873	14.789	21.549	45.211
	lnAGRPRI__2	8.310	11.056	27.042	46.408
	lnAGRPRI_3	8.169	13.310	21.268	42.746

**Table 4 ijerph-20-00384-t004:** Statistical description of coefficients of land variables.

	Variable	Mean	Std.	Min	25th	50th	75th	Max
SCALE	lnSCALE_1	0.000	0.044	−0.341	−0.024	0.001	0.026	0.159
	lnSCALE_2	0.017	0.049	−0.424	−0.008	0.015	0.042	0.227
	lnSCALE_3	0.021	0.042	−0.157	−0.007	0.019	0.047	0.155
ZPG_SCALE	lnZPGSCA_1	0.000	0.044	−0.305	−0.024	0.000	0.026	0.154
	lnZPGSCA_2	0.018	0.049	−0.368	−0.008	0.017	0.044	0.307
	lnZPGSCA_3	0.020	0.042	−0.141	−0.007	0.018	0.045	0.234
AGRE_SCALE	lnAGRSCA_1	0.000	0.025	−0.137	−0.015	0.002	0.016	0.173
	lnAGRSCA_2	0.005	0.025	−0.124	−0.009	0.004	0.018	0.149
	lnAGRSCA_3	0.004	0.024	−0.171	−0.009	0.005	0.016	0.169
PRICE	lnPRICE_1	0.025	0.122	−0.569	−0.031	0.027	0.092	0.449
	lnPRICE_2	−0.013	0.154	−0.738	−0.083	−0.001	0.065	0.490
	lnPRICE_3	0.039	0.173	−0.693	−0.036	0.026	0.118	0.992
ZPG_PRICE	lnZPGPRI_1	0.021	0.143	−0.655	−0.039	0.025	0.087	1.353
	lnZPGPRI_2	−0.013	0.166	−1.501	−0.073	−0.003	0.070	0.489
	lnZPGPRI_3	0.029	0.184	−1.442	−0.048	0.014	0.112	1.222
AGRE_PRICE	lnAGRPRI_1	0.001	0.012	−0.049	−0.006	0.001	0.008	0.052
	lnAGRPRI_2	0.002	0.017	−0.094	−0.006	0.001	0.010	0.148
	lnAGRPRI_3	0.003	0.015	−0.092	−0.004	0.003	0.010	0.093

**Table 5 ijerph-20-00384-t005:** Statistical description of regression results of control variables.

	Variable	Mean	Std.	Min	25th	50th	75th	Max	90%	95%	99%	Sum
SCALE	lnDEN_2	0.043	0.143	−0.273	−0.026	0.035	0.088	1.956	6.268	13.451	39.859	59.577
	lnTER_2	0.046	0.257	−0.799	−0.104	0.042	0.179	1.478	7.183	13.944	37.042	58.169
	lnPGDP_2	0.159	0.192	−0.310	0.058	0.140	0.237	1.842	4.648	10.070	63.310	78.028
	lnFTD_2	−0.032	0.055	−0.202	−0.065	−0.033	0.000	0.498	4.507	9.859	52.042	66.408
	FIN	0.009	0.072	−0.297	−0.027	0.005	0.041	0.391	6.338	14.225	37.183	57.746
	lnPUN	0.000	0.024	−0.142	−0.013	0.000	0.012	0.156	7.465	13.169	31.268	51.901
	lnRAIN	0.056	0.302	−1.669	−0.106	0.070	0.252	1.028	4.507	10.634	49.155	64.296
	lnTEMP	0.648	0.821	−3.973	0.216	0.563	1.139	3.667	5.000	9.014	63.451	77.465
	lnWIND	0.018	0.451	−4.906	−0.197	0.045	0.268	1.594	4.789	10.775	50.634	66.197
PRICE	lnDEN_2	0.066	0.129	−0.454	0.003	0.062	0.111	1.301	7.113	12.183	51.620	70.915
	lnTER_2	0.022	0.249	−0.720	−0.130	0.022	0.139	1.697	6.620	12.042	33.873	52.535
	lnPGDP_2	0.141	0.168	−0.507	0.056	0.127	0.217	1.494	5.423	9.577	60.704	75.704
	lnFTD_2	−0.021	0.058	−0.392	−0.055	−0.023	0.012	0.345	7.254	10.070	48.451	65.775
	FIN	−0.006	0.075	−0.407	−0.031	−0.003	0.030	0.309	8.592	13.310	34.930	56.831
	lnPUN	0.001	0.026	−0.190	−0.013	0.002	0.016	0.135	7.394	13.592	36.408	57.394
	lnRAIN	0.064	0.275	−1.454	−0.085	0.073	0.256	1.057	6.408	10.845	46.408	63.662
	lnTEMP	0.710	0.777	−3.041	0.222	0.610	1.178	3.834	5.282	8.451	62.113	75.845
	lnWIND	0.029	0.437	−4.174	−0.180	0.045	0.263	1.657	5.915	12.746	46.831	65.493

## Data Availability

Not applicable.

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
