# Peer review of "Exploring the Effects of Industrial Land Transfer on Urban Air Quality Using a Geographically and Temporally Weighted Regression Model"

_ijerph, 2022, doi:10.3390/ijerph20010384_

Round 1
Reviewer 1 Report
The manuscript investigates the impact of land transfer behavior on urban AQI in China, with an explicit focus on its spatiotemporal heterogeneity. The paper is solid, well written and organized. I have a few minor suggestions that might be helpful.
1. Sections 2 and 3 can be combined, and some content, e.g., lines 120-168, is repetitive.
2. Temporal effect/change needs a better explanation across section 5.3.
3. Lines 495-512: Policy suggestions should reflect the findings of the four-dimensional analysis, and we can learn from the spatiotemporal heterogeneity.
Author Response
Dear reviewer:
Thank you for your constructive comments on my manuscript. We have carefully considered the suggestion of Reviewer and make some changes. We have tried our best to improve and made some changes in the manuscript.
According to your comments, revision notes are given as follows:
The manuscript investigates the impact of land transfer behavior on urban AQI in China, with an explicit focus on its spatiotemporal heterogeneity. The paper is solid, well written and organized. I have a few minor suggestions that might be helpful.
- Sections 2 and 3 can be combined, and some content, e.g., lines 120-168, is repetitive.
Thanks for your constructive comments, we realized that the structure of this article is not concise and clear enough. So we split the section 3, one part is incorporated into Introduction, and the other part is incorporated into Methodology. And we also revised the lines 120-168 and summarized it into three different impact mechanisms, rather than simply listing the results, so as to avoid repeated narration (in lines 92-100)
- Temporal effect/change needs a better explanation across section 5.3.
Thanks for your constructive comments, we realized that the explanation of temporal change need to be improved. So we conducted supplementary analysis on some temporal changes, such as lines 370-373 and 399-407. However, due to years are not too much in this research, the results have some randomness, especially the price is vulnerable to macroeconomic fluctuations, which makes the price effect more random. Therefore, it is difficult for us to analyze the effects year by year, and we can only try our best to find the main trends or phenomena in the temporal change. The real reasons for the changes need to be explained by subsequent empirical studies.
- Lines 495-512: Policy suggestions should reflect the findings of the four-dimensional analysis, and we can learn from the spatiotemporal heterogeneity.
Thanks for your constructive comments. First of all, we found that the expression of spatial effect was not very appropriate during the modification process, so we deleted this effect and only retained the other three effects. Then, the spatiotemporal heterogeneity is analyzed from these three dimensions. Four conclusions are drawn from these three dimensions: scale effect, price effect, the influence of transfer style on scale effect, and the influence of transfer style on price effect. Finally, in view of the degree and agglomeration of these effects, we have made policy recommendations for developed regions and underdeveloped regions respectively, which is exactly what we get from the spatio-temporal heterogeneity.
Reviewer 2 Report
Based on the four dimensions of industrial land circulation scale, price, mode and spatial effect, this paper uses the air quality index (AQI) and primary land market transaction data of 284 cities in China from 2015~2019 to explore the temporal and spatial heterogeneity of the impact of industrial land circulation on urban air quality. It has high reference value for urban industrial land management decisions based on environmental protection.
Main problems: Slightly confusing structure, unclear methodological statements. Whether the influence of other factors needs to be considered?
Specific questions:
1. The previous statement section is too much, and it is recommended to rewrite Introduction. The Conceptual framework section could be considered for incorporation into Introduction (or Literature review) and Method, respectively
2. Can data and methods be stated separately? Discussions are written separately from conclusions, requiring more in-depth discussions and conclusions. Be clear about how the data has changed and what it is related to.
Ref:Impact of Power on Uneven Development: Evaluating Built-Up Area Changes in Chengdu Based on NPP-VIIRS Images (2015–2019)
L Liu, Z Li, X Fu, X Liu, Z Li, W Zheng
Land 11 (4), 489
3. The specific purpose of the model formula should be distinguished and described in the text, rather than just listing the formula and explanatory parameters.
4. In the summary, lines 1 6 and 18 only mention the magnitude of the impact, not whether it is positive or negative, what does the analysis of "price is more likely to be a tool for local governments to compete for investment" and air quality have to do with it? At the end of the summary, lines 2 0 to 22, return to the topic and state the impact of these recommendations on urban air.
Ref: What matters in the e-commerce era? Modelling and mapping shop rents in Guangzhou, China
X Liu, D Tong, J Huang, W Zheng, M Kong, G Zhou
Land Use Policy 123, 106430
Forecasting Urban Land Use Change Based on Cellular Automata and the PLUS Model
L Xu, X Liu, D Tong, Z Liu, L Yin, W Zheng
Land 11 (5), 652
Statistical analysis of regional air temperature characteristics before and after dam construction
From Chen, From Liu, L Yin, In Zheng
Urban Climate 41, 101085
Smog prediction based on the deep belief-BP neural network model (DBN-BP)
J Tian, Y Liu, W Zheng, L Yin
Urban Climate 41, 101078
5. Please improve the quality of figure, some pictures are not clear enough, the legend is not clear, figure 1 is not specifically described in the text, please outline how to form this framework?
Author Response
Dear reviewer:
Thank you for your constructive comments on my manuscript. We have carefully considered the suggestion of Reviewer and make some changes. We have tried our best to improve and made some changes in the manuscript.
According to your comments, revision notes are given as follows:
Based on the four dimensions of industrial land circulation scale, price, mode and spatial effect, this paper uses the air quality index (AQI) and primary land market transaction data of 284 cities in China from 2015~2019 to explore the temporal and spatial heterogeneity of the impact of industrial land circulation on urban air quality. It has high reference value for urban industrial land management decisions based on environmental protection.
Main problems: Slightly confusing structure, unclear methodological statements. Whether the influence of other factors needs to be considered?
Thanks for your constructive comments, we realized that the structure of this article is not concise and clear enough. We have tried our best to improve and made some big changes on the whole structure. We hope that the revised version will be more readable. Some influence factors have been considered as control variables. However, due to space constraints, there was no in-depth discussion. The significance of these control variables is indeed higher than that of land variables, so they are worth further discussion. In addition, explanation for the heterogeneity also should be research in our future work.
Specific questions:
- The previous statement section is too much, and it is recommended to rewrite Introduction. The Conceptual framework section could be considered for incorporation into Introduction (or Literature review) and Method, respectively
Thanks for your constructive comments. We have simplified some unimportant words in the introduction and directly express the questions to be studied and the contributions of the research. The conceptual framework is also divided into two parts, one is incorporated into Introduction (lines 42-54) and the other in Methodology (lines 188-205)
- Can data and methods be stated separately? Discussions are written separately from conclusions, requiring more in-depth discussions and conclusions. Be clear about how the data has changed and what it is related to.
Thanks for your constructive comments. Referring to the paper you recommended, we revised the structure of the Methodology. We introduced the data source, analysis framework and model in turn, and redraw the framework diagram to illustrate the data processing process and analysis ideas. The discussion and conclusion are also separated and rewritten. The discussion has a more in-depth content, including the main findings of this article, the supplement and inspiration to the existing literature, limitations and future research directions. The conclusion part reorganizes according to the new analysis framework, which includes scale effect, price effect, the influence of transfer style on scale effect, and the influence of transfer style on price effect. The discussion and conclusion may be more readable than before.
Ref:Impact of Power on Uneven Development: Evaluating Built-Up Area Changes in Chengdu Based on NPP-VIIRS Images (2015–2019)
L Liu, Z Li, X Fu, X Liu, Z Li, W Zheng
Land 11 (4), 489
- The specific purpose of the model formula should be distinguished and described in the text, rather than just listing the formula and explanatory parameters.
Thanks for your constructive comments. After the introduction of the model principle, we introduced the specific form of the model used in this study. It explains the regression strategy and what the regression results focus on (lines 266-286).
- In the summary, lines 1 6 and 18 only mention the magnitude of the impact, not whether it is positive or negative, what does the analysis of "price is more likely to be a tool for local governments to compete for investment" and air quality have to do with it? At the end of the summary, lines 2 0 to 22, return to the topic and state the impact of these recommendations on urban air.
Thanks for your constructive comments and references. Whether the effect is positive or negative, the price effect is always greater than the scale effect. Based on previous studies, we speculate that this may be because the construction land indicators are strictly controlled, resulting in relatively stable spatio-temporal changes in the transfer scale between different years, so the interpretation of urban air is relatively small. Further, we speculate that local governments are more likely to reduce land prices rather than expand land scale to attract investment, thus reducing air quality. Because the length of the abstract is limited, our statement is too brief and results in ambiguity. In this regard, we have revised the relevant statements in the summary. In addition, lines 20-22 of the summary have also been revised, emphasizing that the starting point of the policy recommendations is the spatial heterogeneity of environmental effects, and its goal is to balance the relationship among industrial land, industrial development and air quality.
Ref: What matters in the e-commerce era? Modelling and mapping shop rents in Guangzhou, China
X Liu, D Tong, J Huang, W Zheng, M Kong, G Zhou
Land Use Policy 123, 106430
Forecasting Urban Land Use Change Based on Cellular Automata and the PLUS Model
L Xu, X Liu, D Tong, Z Liu, L Yin, W Zheng
Land 11 (5), 652
Statistical analysis of regional air temperature characteristics before and after dam construction
From Chen, From Liu, L Yin, In Zheng
Urban Climate 41, 101085
Smog prediction based on the deep belief-BP neural network model (DBN-BP)
J Tian, Y Liu, W Zheng, L Yin
Urban Climate 41, 101078
- Please improve the quality of figure, some pictures are not clear enough, the legend is not clear, figure 1 is not specifically described in the text, please outline how to form this framework?
Thanks for your constructive comments. We revised the research framework. It is found that the expression of spatial effect is not very appropriate, so this effect is deleted and only the other three effects are retained. Therefore, Figure 1 has also been modified and its clarity has been improved, and its related expressions have been improved. See lines for details. The original image file of other pictures is too large. If the clarity is reduced when embedding in word file, we can provide additional high-definition original images. The legend is generated by Natural breaks classification, and its advantages and disadvantages are supplemented in the article(lines 325-328)